# Improving Fundamental Movement Skills during Early Childhood: An Intervention Mapping Approach

**DOI:** 10.3390/children10061004

**Published:** 2023-06-02

**Authors:** Alexandra Patricia Dobell, Mark A. Faghy, Andy Pringle, Clare M. P. Roscoe

**Affiliations:** 1Institute of Applied Health Research, School of Medical and Dental Sciences, University of Birmingham, Edgbaston, Birmingham B15 2TT, UK; 2Human Sciences Research Centre, College of Science and Engineering, University of Derby, Kedleston Road, Derby DE22 1GB, UK

**Keywords:** fundamental movement skills, physical activity, early childhood, intervention mapping

## Abstract

The earlier in life that a child can begin mastering fundamental movement skills (FMS), the more positive their physical activity (PA) trajectories and health outcomes are. To achieve sufficient development in FMS, children must be guided with tuition and practice opportunities. Schools and educators provide an opportunity for interventions that improve health behaviours and outcomes for children. The aim of this study was to use intervention mapping (IM) to design a programme of school-based intervention to improve FMS for children aged 4–5 years old. Following the six steps of IM, with each step comprising three to five tasks that require the input of a planning group formed by key stakeholders, a programme of intervention was planned. Prior knowledge and primary and secondary evidence was used to support the development of the programme. A logic model of the problem as well as logic models of change, programme design, production, implementation, and evaluation were proposed or completed within the study. The results can be used to begin to implement an FMS-focussed intervention within school settings within England and propose a sustainable and realistic approach for helping children to develop FMS with the support of well-informed educators who are confident to deliver better FMS practice and PA opportunities.

## 1. Introduction

### 1.1. Physical Activity and Fundamental Movement Skill Interventions during Early Childhood

There is an abundance of studies in the literature supporting the relationship between FMS competency and PA throughout childhood [1,2]. The earlier a child can begin mastering FMS through appropriate tuition and practice opportunities [3], the more positive their PA trajectories [4] and health outcomes, including adiposity and motor development, will be [5]. Despite this evidence, during the early years, research should focus on both FMS competency and performance of PA as separate elements of a child’s health behaviours. The literature during early childhood shows that the relationship between FMS and PA is weak [6]; despite this, the relationship is seen to strengthen into middle childhood and adolescence [2]. This provides a strong basis and argument for the dedicated provision of FMS practice and PA opportunities for a young population to promote and instil healthy behaviours and habits.

Interventions to increase FMS competency in children are common [1,7,8,9], specifically during early childhood [10,11]. However, many interventions fail to use in-depth planning and mapping procedures to create interventions that can be implemented with long-term, sustainable approaches and that are based on the needs of key stakeholders [12]. This results in the benefits of interventions being short-lived and lacking longitudinal follow-ups, which brings into question the lasting effectiveness and sustainability of these programmes, a concern expressed by early years educators in England [13]. Despite these issues, recent changes in the research landscape have been documented. Daly-Smith et al. [14] evidenced a whole-school approach to increasing the PA levels of children, using a multi-stakeholder experience, a contrasting approach to previous research of academic and external-provider delivery approaches to interventions. Goss and colleagues [15] also provided a strong argument for the involvement of multiple stakeholders within the development of assessing physical literacy in schools, collecting key views and opinions on how to ensure that physical literacy development can be achieved in these settings. Other methods recently implemented to improve the planning, production, uptake, and sustainability of interventions include collective intelligence [16,17]. Collective intelligence is an applied systems science approach to establish structural models of the problem and structural models for a solution. Most importantly, this research engaged key stakeholders of interventions, including researchers with FMS intervention implementation experience, teachers, coaches, and finally, public health specialists, which mirrors that of intervention mapping (IM). Ma et al. [16,17] identified barriers to intervention implementation and thoughts surrounding possible solutions, and they provided rich data to help inform programmes of change through effective planning. These studies demonstrate the growing evidence for involving key stakeholders when designing interventions using an iterative approach, and in doing so, they highlight the multiple methodological designs and applications that facilitate inclusivity and engagement among stakeholders in the research processes. Key stakeholders may include the direct recipients of the intervention; however, it is essential to consider the individuals who enable the interventions to take place, known as the implementers, adopters, and maintainers [12]. In the case of this research, children, teachers, parents, coaches, and researchers are key in the pivotal developments of novel, effective, and sustainable interventions.

It is recognised that behaviour change, public health, and PA intervention is complex, multifaced, and interactive [12,18]. Although not all determinants and components of health can be addressed within a single intervention or even a programme of interventions, the most important influences must be addressed and utilised. For example, in a feasibility trial, Langford et al. [19] found that a parental web component of a PA intervention for pre-schoolers in the UK was ineffective, with a low number of interactions present in this element of this intervention. Parental involvement is important to increasing the PA behaviours of young children, and further methods should be explored and utilised to ensure the improved engagement of parents to reach programme goals. This example highlights the importance of considering the wants, needs, and participation of key stakeholders in the initial processes and design while allowing periods of trialling and testing feasibility. Using this iterative design for intervention development processes leads to programme adaptations that allow for the best possible outcomes for the end users while remaining cost-effective at a public health level [20]. Iterative and cyclical review processes are essential to long-term programme design, allowing for a ‘back to the drawing-board’ approach and for the application of different theories and practical applications from the literature within intervention design [12]. 

As mentioned, health promotion interventions must consider the target population, their physical and social environment determinants, and as many influential determinants of behaviour as possible. For children, these factors range from family influence [21,22], deprivation, sex, and ethnicity [23,24,25,26,27], which were highlighted in a recent systematic review [24], in addition to age, stage, and maturation status [28]. Educational environments have been considered important spaces for both PA attainment and intervention in child populations, and for some children, this may be the only environment where they attain good-quality PA [29,30]. Primary education settings in England have access to ring-fenced funding for physical education (PE) and sport [31]. Importantly, schools have autonomy over how they decide to spend their funding, including resources such as equipment and external coaching, and to fund continuous professional development and training for primary educators. Funds can be used to increase educators’ skills, knowledge, confidence, and the use of appropriate pedagogical practices within PE. However, there is little conversation involving the educators themselves regarding their development needs. The need for better preparation of educators to teach PE and facilitate PA has been cited as an issue multiple times despite this funding [13,15,32,33]. Despite a clear need for more effective guidance for PE, PA, and early childhood physical development, there is not only little statutory provision for the educators of children but also a lack of approaches that join up schools with communities with the aim of improving the FMS of children as well as enhancing their PA. A recent expert statement suggests that policy-level recognition of motor competency, including FMS development, needs to be provided so that adequate funding is made available to enhance the skills of practitioners inside and outside of the school environment [34]. In making the case for effective provision, strengthening the proposal to policymakers and governments is key. Health-based policy has been criticised for not considering the wider determinants of health behaviours, not involving key stakeholders [12], and its need for processes that better consider this complexity in intervention design. To this end, the process of IM provides the collection of key evidence and theory for designing and implementing effective intervention that address the multi-layered determinants identified by key stakeholders, and it involves them in an iterative process to shape solutions that best address their health needs [12]. 

### 1.2. Introduction to Intervention Mapping 

Intervention mapping was initially designed to be used in health promotion interventions and programmes [12]. It is underpinned by a number of key principles, including the participation of key stakeholders, bottom-up and iterative approaches, consideration of wider influences and determinants on behaviour, underpinning theory, and an aspiration to intervene [35]. IM is composed of six key steps. These six steps are made up of related tasks (Figure 1) to complete the mapping process and result in the creation of a programme of effective, sustainable interventions [36]. As a reflection of the principle of participation, a planning group of key stakeholders is involved throughout the IM process to inform its development. In Step 1, existing evidence is crucial to understanding the problem that an intervention aims to solve and address. The existing literature examined within this introduction and unpublished data collected in relation to this research study help to establish the problem within the population, informing the logic model of the problem. This evidence also helps to establish what needs to be changed through individual behaviours in addition to changes within the environment, which ultimately inform the logic model of change established in Step 2, which is composed of programme outcomes and objectives. In Steps 3–5, programme design, production, and implementation form the mapping process, and these components should be informed by key stakeholder opinions and experience, including qualitative data sets [13]. Step 6 of the mapping process allows an evaluation plan to be established. This plan should consider practical and meaningful ways to evaluate the effectiveness and the need for improvement of a programme. The design of this plan should be informed by pre-existing reliable and valid measurements [24] used to measure the desired population. 

Reflecting an understanding of the wider influences on health, intervention mapping is heavily influenced and informed by Bronfenbrenner’s [37] socioecological model (SEM; Figure 2), which highlights that an individual’s behaviour is influenced by several different determinants that exist at different levels and interact between these levels. When considering a child’s FMS competency and PA levels, we can consider determinants using the layers of the SEM.
Individual determinants are examined, including a child’s beliefs and attitude towards PA in addition to their enjoyment of activity. These determinants can be affected by a child’s immediate interpersonal environmental determinants, including members of a child’s family, their friends, and their peers, with knowledge and belief being shared between these groups. Organisational and institutional determinants are highly influential for children, especially in the early years. All children in England can experience the school setting, and therefore, it remains a key determinant influencing children’s choices and behaviours. A sense of community can help to foster better health behaviours. Stronger relationships between parents and schools, in addition to efforts by local authorities, can determine the health of a community. The highest-level determinant is public policy. When this determinant is considered, examining the policy for early PA and PE, in addition to dissemination of policy and statutory training of practitioners are key determinants. It is therefore important that programmes of intervention focus beyond the end recipient and includes key stakeholders.


Combining what the authors of this study know about FMS and PA, interventions during early childhood, and the school environment, these three elements should be considered to be important for early childhood health and PA. As aforementioned, there have been multiple interventions at the school level, and although IM has been used to plan interventions to combat obesity in early childhood populations [38,39,40], to the authors’ knowledge, there are no studies to date that have used the IM approach proposed by Bartholomew-Eldredge et al. [12] to increase FMS competency in early childhood, and as such, this research is a novel contribution to the field.

### 1.3. This Research

Considering the evidence introduced here, the authors believe IM proposes a suitable and robust method to plan interventions, having been recommended as a valuable tool for early childhood intervention development [41]. The aim of this study was to present a set of initial IM process and outcomes for developing effective FMS interventions within schools for the early years age group as the ‘first-step’ in several processes leading to optimal and sustainable outcomes for intervention programmes. IM is used as a framework to structure and guide the approach in this study.

## 2. Methods and Results—Six Steps

This study followed a non-traditional IM format, detailing the initial stages of IM development with scope for further study consideration. This means that the researchers were pragmatic and practical when developing this IM iteration. This article uses multiple perspectives based upon the SEM, focussing on the child at the centre (individual), their parents as key interpersonal influences, teachers at the organisational level, and researchers who could be considered to be on both the community and national levels, depending on their work and impact. Data was taken from multiple sources, including the authors’ own work, to identify the most important determinants to these children. The authors also used these multiple perspectives and sets of data to shape the solutions that are proposed throughout this article, showing a strong alignment with IM principles, including participation, a bottom-up approach, and intervention. Sharing the information from IM development at formative and summative phases on process and impact are important in order for others to gain from this work, and as recommended by the IM workbook, publishing work before the end of the process is a valuable process [42]. Further depth and explanation on how these goals were achieved are given appropriately within each step.

### 2.1. Study Design

Using the six-step process of IM (Figure 1) proposed by Bartholomew-Eldredge and colleagues [12], each step and series of related tasks was adopted as a framework to guide the research, and when referring to IM, Bartholomew-Eldredge et al. [12] are acknowledged. IM programmes are designed using a strong basis in theory, evidence, and stakeholder holder involvement. This requires the formation of a ‘planning group’ made up of key stakeholders to the programme to shape the development of interventions in a way that is needs-led and practical, and the members highlight and guide the choice of key theories and evidence to inform the programme while relating the goals and aims of the programme to real-life situations and outcomes. The planning group’s contribution is pivotal in achieving the sustainable nature of the individual interventions and whole programmes proposed through this method. 

The six-step model and its associated tasks are further explained in the following sections; because IM is an iterative process, the authors discuss the steps/tasks combined with the results from this study that emerged from their deployment, and this helps preserve the context in a way that would not be possible if the method and the results were separated. Each section states how each task within each step of the mapping process was completed and how the planning group was involved. Although further involvement of the planning group may have been advantageous at various stages of this study’s process, this was not always realistic for the lead researcher and the members of the planning group. The authors used a more pragmatic approach by using several initial focus group sessions within Step 1, with the planning group split into respective groups (children, parents, researchers, and educators) where necessary. Proposed questions for each step of the planning process were discussed at this first step (and can be viewed in the Appendix A) of the IM process, and the results are detailed in each respective section. If there was a need for further elaboration or information from the planning group, individuals were contacted separately via email or telephone during the later stages of the IM process. This approach was used to appropriately reduce the participants’ burden.

This study displays a strong mixed-methods design, with the use of a planning group to convey their needs and opinions in addition to the use of the data collected in previous studies [13,24], including currently unpublished empirical quantitative data. Mixing data in the planning, collection, analysis, reporting, and interpretation stages of this study was essential, and the mixed-methods approach to the reporting of findings and the future assessment of the IM process are also demonstrated, with numerical programme outcomes and qualitative discussions with key stakeholders. The current approach provides a comprehensive, participant-informed, and informative account facilitated via the IM process [12].

Ethical approval (ETH2021-3572) was gained from the University of Derby Science and Engineering Ethics Committee, and all participants completed written and verbal assent before involvement in any stage of the IM process.

### 2.2. Step 1: Logic Model of the Problem

As shown in Figure 1, Step 1 comprised four steps. A planning group was established, followed by a needs assessment and the production of a logic model of the problem. Subsequently, the programme context, population, setting, and community were identified, and finally, the broad programme goals were stated. 

#### 2.2.1. Task 1.1: Establish the Planning Group

The planning group included representatives of the target population—i.e., children—and environmental agents and programme implementers—i.e., teachers, members of school senior leadership teams, parents, and PA and motor competence researchers. Other stakeholders to consider in further IM iterations include local authority representatives and community coaches.

Participants of the planning group were recruited through word of mouth, social media, contacts from previous research studies, and outreach via email by the lead researcher, and where necessary, gatekeeper approval to work with children in school settings was gained. The planning group was considered to be trustworthy and knowledgeable in the subject matter. Children who participated in this study were recruited from two schools in the central area of England. As a preliminary and small-scale mapping process, it was concluded that samples should reflect critical mass and representativeness, and therefore, for every SLT member, there should be 3–4 early years foundation stage (EYFS) teachers (represented in Dobell et al. [13]), and for each EYFS teacher, there should be around 10 children. The parents of the children recruited were also invited to participate, in addition to parents who expressed an interest through social media or word of mouth. The invited researchers were invited by the lead researcher to participate and had pre-established connections through institutional relationships or networking. 

Prior to their involvement, participants were provided with a participant information sheet and the opportunity to contact the lead researcher to ask any questions they had about the research project before consenting to take part. For child participants, consent was first gained from the school headteacher, and subsequently, parents were asked to consent to their child’s participation within the study. Verbal consent was gained from the children at each visit.

##### Planning Group Protocol and Involvement

Planning group focus groups were used to explore and to execute several of the tasks in Steps 1–6; however, the focus group protocols are explained in Step 1. The lead researcher felt it was important to engage with the stakeholders in multiple ways to ensure all views could be expressed by both the group and as individuals and to reduce participant burden by splitting participation into smaller activities. The lead researcher held regular discussions with all members of the research team at each stage of the process, ensuring reliable and valid research outcomes. 

Three focus groups took place following a semi-structured approach using a proposed schedule of questions covering areas for Step 1–6 (see Appendix A); the main aim was to facilitate discussion between members of the planning group and establish the key and prominent themes discussed by the members. The focus groups were recorded and transcribed verbatim. Focus group discussion followed a constructionist research paradigm which argues that there are multiple realities that create a social reality. This is especially true of this study due to the number of stakeholders involved and how they each view early childhood PA and practice.

Two adult focus groups were formed. Group one consisted of four parents of 4–5-year-olds (FG1P). Group two consisted of three PA and MC researchers (FG1R). The final sub-group of the planning group comprised two groups of children aged 4–5 years of age (FG1C). Although key to the planning group, specific focus groups were not held with teachers, coaches, or school senior leadership members. This was deemed appropriate, as previous work by the authors collected a wealth of information related to IM from these key stakeholders (see [13]). Where focus groups did not seem appropriate to the context, other means, including written feedback sheets, were administered.

Traditional, formal focus groups were deemed unsuitable for young children; therefore, during their focus group, children completed a ‘write, draw, show and tell’ task [43] to facilitate the expression of young children’s feelings and ideas. To inform Step 1, children were asked to ‘write or draw about what you enjoy about PE’. Following the drawing and/or writing, the children shared their work with the rest of the group and told them about what they had produced. Children were further prompted by the researcher or the class teacher to give as full answers as possible. The works the children produced were each photographed and assigned a participant code (Appendix A). 

#### 2.2.2. Task 1.2: Needs Assessment and Logic Model of the Problem

Within the second task of Step 1, a needs assessment of the literature and existing knowledge was conducted to establish the problem, whose problem it is, who the problem was affecting, what behaviours and environmental conditions are causing or are related to the problem, and the determinants of these behaviours and the environment (Table 1). First, the problem was stated: **Problem:** children are not sufficiently competent at FMS in the early years. This negatively affects their levels of PA, quality of PA, and opportunities for PA social interactions; this leads to unhealthy PA habits developing, causing poor health outcomes such as obesity.

The planning group was consulted during the focus group discussions about their thoughts and ideas for areas that had been missed, overlooked, or that should be added into the needs assessment. A summary of themes and comments made in the focus groups relating to Step 1 can be found in Appendix A. This table also provides key comments and quotes from the interviews undertaken in Dobell et al. [13] study to support Step 1. 

Following the focus group discussions and answering of questions appropriate to Step 1, a final logic model of the problem (Figure 3) was created; this model helps to summarise the sequence used and information established in task 1.2. Reflecting the wider influences on health, the logic model of the problem shows the personal determinants that children face (Phase 4) and the personal determinants that key environmental agents (teachers, etc.) face (Phase 4). These determinants contribute to the environmental conditions that impact the PA/FMS behaviour of children (Phase 3), which contributes to health problems (Phase 2) and children’s quality of life (Phase 1).

#### 2.2.3. Task 1.3: Programme Context, Population, Community, and Setting

The planning group was key in describing the context and setting of the interventions as well as key population characteristics and needs. The information provided during the focus groups, which was expressed in a study by Dobell et al. [13], researcher knowledge, and published literature, allowed an asset assessment to be undertaken (Table 2). The social, information, policy/practice, and physical environments’ assets to aid the success of a sustainable intervention were assessed and stated. Although each asset assessment element presented four areas to target, just one example for each area was chosen (Social: primary schools; Information: school newsletters/bulletins; Policy/Practice: teacher training for early years (school level during this intervention); Physical Environment: school spaces). The linked-up approaches of interventions within a programme must consider each asset assessment element fully. All areas of each element of the asset assessment should be considered to be critical, and the iterative process of IM allows for future development within each area. 

#### 2.2.4. Task 1.4: Broad Programme Goals

The final task of Step 1 was to state the broad programme goals, i.e., what the programme hopes to achieve overall. Using the logic model of the problem, broad outcomes to be changed through the implementation and maintenance of the intervention programme were stated: (1) increase the FMS competency of children in EYFS; (2) increase the quality of PA of children in EYFS; (3) improve the quality of FMS tuition/guidance.

### 2.3. Step 2: Programme Outcomes and Objectives—The Logic Model of Change 

Step 2 comprised five tasks (Figure 1). The expected outcomes for changes in behaviour and the environment were stated, followed by the performance objectives to achieve these behavioural and environmental outcomes, complimented by the selection of the most important determinants for these behavioural and environmental outcomes. Task 4 created the matrices of change, establishing change objectives for each determinant and leading to the logic models of change being formed and presented. 

#### 2.3.1. Task 2.1: Expected Outcomes for Behaviour and the Environment

Following the production of the logic model of the problem, expected behavioural and environmental outcomes for the programme of change were stated. Programme outcomes are the desired changes to be made by implementing the programme, leading to the overall broader programme goals. By ‘doing the flip’ and moving from focussing on the problem to focussing on the solution, a logic model of change can be formed. Behaviours causing poor health and health problems from the logic model of the problem were ‘flipped’ to reveal several desirable and health-promoting behaviours, and a selection of the most important behaviours and environmental conditions was made based on the literature and previous knowledge. A list of the potential desired outcomes for the target population’s behaviours and environmental outcomes were stated (Appendix A Appendix A). NICE [44] recommend that brief advice is used in intervention outcomes; therefore, within this programme, two main outcomes were stated and taken forward in the next steps. 

**Behavioural:** 1.0 Increase the number of EYFS children engaging in FMS practice at school.**Environmental:** 2.0 Increase the provision of FMS delivery by 20% at the EYFS in schools.

Engaging children in more FMS practice was identified as key. When a cohort of 4–5-year-old children (unpublished data) were examined, most were meeting the PA guidelines, but many children had low FMS competency. This made a clear case to focus on improving FMS over simply improving the children’s PA levels. Second, improving the provisions of FMS delivery in school was also revealed as a key outcome. Although teachers feel that there are enough PA opportunities in schools [13], there was identification of a lack of support to implement FMS, and there was some suggestion that the EYFS framework lacked guidance to provide the adequate provisions. 

#### 2.3.2. Task 2.2: Performance Objectives for Behavioural and Environmental Outcomes

To reach the desired programme outcomes, performance objectives for the population behaviours and their environment were established. These performance objectives are specific sub-behaviours for the children’s health-promoting behaviours; if the children are to perform more FMS practice, they must first engage in these sub-behaviours.


**Performance Objectives for Behavioural Outcomes:**
**1.1.** Increase the percentage of children spending 10 % more time doing PA in school each week**1.2.** Increase the percentage of the structured and unstructured FMS-related activities a child partakes in each week by 25%**1.3.** Increase the number of goals set for children’s FMS and PA performance by two each term


The following performance objectives are specific sub-environmental actions for teachers/senior leadership teams (SLT) to engage with to change the environmental conditions so that schools can be better prepared to deliver further FMS-related activities.


**Performance Objectives for Environmental Outcomes**


**2.1.** Increase the number of teachers facilitating FMS practice in EYFS settings**2.2.** Increase the number of teachers planning for PE, PA, and FMS in schools**2.3.** Increase the number of teachers aware of the benefits of providing EYFS children with PA opportunities

#### 2.3.3. Task 2.3: Determinants for Behavioural and Environmental Outcomes

Behaviours are formed by determinants of the individual, whereas environmental conditions are formed by the determinants within the environment which impact people. These determine if a child or teacher (environmental agent) will complete a performance objective (behaviour). Determinants need to be targeted to increase the likelihood of change for the individual and their environment. The determinants for the individual and health-promoting behaviours included: mastery level of FMS (skills), self-efficacy and physical self-concept, perceived norms, knowledge, fitness level, and sedentary behaviour. The determinants for environment and agents within them included: knowledge, self-efficacy, social norms, parental beliefs, outcome expectations of the setting, and the attitude of the school to PA, physical skills, and health.

The most influential determinants were selected from the previous lists according to how important they were and how changeable they could be with intervention, according to (i) research conducted by the authors ([13,24], unpublished data), (ii) prior knowledge, and (iii) the literature. For example, self-efficacy is largely reported as an important element of PA as children age [45]; therefore, it was deemed important to focus on when developing this intervention. Levels of FMS competency have also been previously positively influenced through shorter term intervention [8]; therefore, we know FMS competency can change for children through intervention, although longer-term effects have not been demonstrated as successfully. This gives scope for an intervention with a longer-term approach (e.g., over a whole school year) to be developed, keeping in mind individual behaviour changes while targeting environmental agent determinants to ensure better sustainability. 

#### 2.3.4. Task 2.4: Create Matrices of Change

This task involved creating matrices or tables. This was achieved when each determinant chosen in Task 2.3 was crossed with a performance objective (Task 2.2) to create change objectives. Using this information, matrices of change objectives were constructed, bringing all the tasks in Step 2 into a consolidated place (please see Appendix A). To ensure the matrices and chosen objectives were relevant to the key stakeholders, the planning group was asked to review these. A written feedback task, which included rating individual behavioural and environmental characteristics (determinants) in addition to perspectives on how achievable the change objectives were perceived to be, was assigned, and the results were gathered. All adult participants (n = 7) were contacted to provide written feedback, and 71% (n = 5) returned this information to the lead researcher. 

#### 2.3.5. Task 2.5: Logic Models of Change

Using the performance objectives and behavioural and environmental outcomes established in this step, the three logic models of change were proposed (Figure 4, Figure 5 and Figure 6).

### 2.4. Step 3: Programme Design

Step 3 comprised three tasks. First, the authors generated the programme’s themes, components, scope, and sequence. This was followed by establishing the theory and evidence-based change methods of the programme and, finally, the applied practical applications to achieve change in the interventions. 

#### 2.4.1. Task 3.1: Programme Themes, Components, Scope, and Sequence

In this task, the theme, components, scope, setting, and sequence of the programme of interventions were generated. These elements of IM require planning group input as well as researcher knowledge created within the wider research project [13,24] (unpublished data) and existing literature (what has and has not worked in the past) [11,46,47]. Within the initial focus groups (FG1P, FG1R, and FG1C), the second section of discussion centred around the key elements to inform Step 3. Questions were component- and theme-centred, including ‘What characters/books do you find engage children in learning’, and ‘When designing a multicomponent intervention, what is the maximum number of components you would suggest using’. The information gathered from these focus groups and interviews from Dobell et al. [13] are shown in Appendix A. The following sections explain how the theme, components, scope, and sequence should achieve successful delivery of the programme. 

The theme is a general organising construct for the programme that usually relates to the change objectives and intervention environment, and as such, this programme’s theme was EYFS FMS in school. This established the simple name of the programme, ‘The FMS School Project’, making the components of the interventions recognisable. Through stakeholder engagement, additional themes to engage children were established as important; therefore, the framework created was developed using suggestions of themes for activities to inspire children’s engagement, including animals, superheroes, and book characters (Appendix A).

The components make up the body of the intervention and must be strongly related to the performance and change objectives identified in Step 2 (Table 3). This programme was made up of two main components/interventions, which are related to one another and will be discussed in further detail in Step 4. The chosen methods and practical applications (see Section 2.4.2) involved within each component of the intervention are detailed in Task 3.3, and an outline is provided in Table 4. A framework intervention, which describes and guides teachers to improve children’s FMS with the autonomy of the intervention structure and implementation activities, was chosen over an intervention of strict set session deliveries for teachers to follow. Previously, educators mentioned the need for adaptability in delivery; a framework helps to demonstrate how sessions can work in the chosen setting but allow the flexibility of adaptation to individual classes and abilities [13]. Unpublished observational data evidenced that children across a single EYFS class will likely need the adaptability that a framework intervention provides. The researcher focus group (FGR1) also highlighted the need for teachers to have ownership of the intervention to improve implementation, longevity, and sustainability. 

Within IM, the intervention scope must be realistic not only to meet the goal of being sustainable for the implementers and maintainers, but also for use by the individuals and environmental agents. When the previous literature was explored, it was found that it is common for school-based FMS interventions to last over a term (10 weeks) and to be delivered by an individual external to the school (researchers/coaches). Pedological literature continues to argue that delivery by teachers that work with children on a continued and regular basis is important not only for development of skills, but also for the relationships between children and teachers [48]. This method also helps to promote teacher education, knowledge, and self-efficacy. Therefore, it was important to this programme of intervention to ensure the delivery of intervention two (framework) was by teachers. The scope of this programme of interventions includes the delivery of the intervention to teachers (training) and the delivery of the intervention by teachers within schools (framework). The scope considers who delivers the intervention, how it is delivered, the setting of delivery, how long it is delivered, and the evaluation of its elements (Table 3).

The sequence of the intervention relates to the order of how the components should be delivered. This intervention is made up of two key components, and one must be delivered before the other can occur: teacher training by delivery partners, followed by a school-based framework implementation delivery by teachers at the EYFS. These would both require evaluation following their delivery as carried out by the implementer (see Step 5). 

#### 2.4.2. Task 3.2: Theory and Evidence-Based Change Methods Chosen

Following Task 3.1, initial theory and evidence-based change methods that were well-suited to the ideas generated through the previous steps were chosen to deliver the programme of interventions. Change methods are rooted in behavioural theories and psychological principles and are defined as general processes for influencing change at individual and environmental levels and the determinants within these. Kok et al. [49] specifically developed a taxonomy of behaviour change methods for IM to help to guide the decisions made within this task. The definitions and parameters provided help with framing the scope of each method and deciding whether it would fit well with the components, scope, and sequence (Task 3.1). Change methods were split into categories to address different areas of the problem. Using basic methods to begin with, to identify the broad ways to create change for a school-based intervention, these methods were matched with practical applications (Task 3.3). The chosen change methods are shown in Table 4, which also presents the broad practical applications chosen.

#### 2.4.3. Task 3.3: Practical Applications to Achieve Change in Intervention

The final task of Step 3 was to start to design practical applications, practical techniques that operationalise the change methods chosen, to be used within the individual interventions of the programme, fitting within the intervention group(s) and the context. Therefore, this task collated information provided by the planning group, the Dobell et al. [13] practitioner interviews, and the written feedback exercises (Appendix A) with the performance outcomes and change objectives within the logic models of change (Step 2). Regularly suggested applications by the planning group and successful applications within the literature were considered within the programme of intervention context. For example, multiple educators have mentioned the need for adaptability and to build from a ‘framework’ to provide teachers with structure [13]; therefore, it was deemed a possibly appropriate method. In Table 4, the basic behaviour change methods and practical applications according to the determinants of the problem can be seen. The parameters of these are also stated. The adaptable framework and training for teachers were then further developed by choosing behaviour change methods from Table 4 and developing these via the specific and detailed applications as intervention elements (Table 5). Providing detailed change methods and applications in addition to the population, context, and parameters of each method were stated to show how implementation could occur within EYFS settings.

### 2.5. Step 4: Programme Production

In Step 4, there were four tasks (Figure 1). Task 4.1 refined the programme’s structure and organisation of interventions, followed by the preparation of the plans for the programme’s materials. Subsequently, the programme’s message, materials, and protocols were drafted, and finally, the materials drafted in that task were produced, pre-tested, and refined.

#### 2.5.1. Task 4.1: Refine Programme Structure and Organisation

Within Step 4, the aim was to produce an effective programme by refining the structure and organisation of the theoretical change methods and applications that were proposed in Step 3. This step organised how these systems of change would be delivered within the programme itself and the chosen implementation environments. The application created at this stage described the theme, scope, sequence of delivery, and delivery channels/vehicles of this programme. At the highest level, this programme has two main interventions: teacher training and the delivery of an adaptable framework for children. The interventions were made up of multiple constituent parts (applications) to achieve the programme’s goals, which include the behavioural and environmental outcomes, performance objectives, and change objectives that were established in the previous IM steps. Within Table 6, the system of delivery, who delivers each part, and how and when they are delivered can be seen; they are integral to the organisation and structure of the programme. This includes channels of delivery and communication (interpersonal or mediated), which are key for disseminating the information about the intervention and delivering it effectively. Appendix A also represents the overall structure and organisation of the programme, demonstrating where practical applications should sit. 

#### 2.5.2. Task 4.2: Prepare Plans for Programme Materials

Using the information gathered in the prior IM steps and from the planning group’s input (Appendix A), plans for the materials were produced. Full working documents were produced for a framework booklet (FMS information practical sessions, ideas for classroom sessions, evaluation, and assessment) and one of three training session slides, and they will be further described in the following sections (these are available upon request from the corresponding author A.P.D.). Specifications and working documents were not produced for all applications within this study in order to be pragmatic in the approach and development of future research (this research was completed as part of a three-year PhD project). However, descriptions for these future specifications were made and would include promotion videos of the intervention, local authority communications with schools, additional parental communications, more in-depth classroom activity approaches, and websites for sharing information. Table 7 presents all of the programme’s components, their descriptions, and the producers needed for these. Collaboration with other experts, e.g., media producers and website creators, would be essential in creating an intervention of high quality that is sustainable and will have a meaningful impact. 

#### 2.5.3. Task 4.3: Draft Message, Materials, and Protocols

In the next task, the plans to create the components were put into practice (and this process would need to occur for the remaining elements in Table 7 during future research). The overall message of the programme was to increase FMS and PA through structure and fun during the early years with a method designed to engage the participants, implementers, adopters, and maintainers of the programme, with this key message being strongly informed by the focus group sessions and by the Dobell et al. [13] interviews. As the theme centred around improving FMS for EYFS children and the setting was schools, the programme was named ‘The FMS Schools Project’; the logo can be seen in Appendix A.

Using this central theme, the materials and protocols to support the practical applications chosen in Step 3 and plans made in Task 4.2 were drafted, including a framework booklet document and training sessions for teachers. The booklet is a key material for both the training intervention and the delivery of the school intervention. It is divided into sections framing key intervention elements and written with teachers in mind by presenting the evidence from research and the literature in a non-specialist and consumable way. 

The first section helps teachers to strengthen their understanding of FMS, PA, and health by introducing and reinforcing what FMS are by visually splitting them into their domains (locomotor, object control, and stability). This is further developed by acknowledging how FMS should be part of PE for EYFS children while linking back to health and academic outcomes for children. The second section of the booklet focusses on how to plan for FMS tuition in school and how teachers can improve and promote it within the EYFS as well as on key delivery methods. This information within these two sections compliments the first session of training, which explores why FMS and PA are important for children’s health and how the school environment can help to improve these outcomes for children. In the practical workshop setting, this is achieved by first surveying the teacher’s current school environment and what they already do well before exploring where improvements could be made.

The third section of the booklet provides an array of practical activity suggestions, with many being informed by the children’s preferences presented in FG1C (Appendix A and Appendix A). The practical sessions are split by specific FMS domains: locomotor, object control, and stability, or a combination of skills; this is followed by how to gain formative feedback with children. Throughout the focus groups (FG1R and FG1P) and previous interviews [13], there was a clear suggestion that activities should occur outside of allotted physically active times. For example, this could be done by linking another curricular area such as maths or literacy to FMS learning. The fourth section focusses on these opportunities by using techniques to increase stimulus and knowledge around FMS. These sections of the booklet compliment session two of the training intervention, which introduces teachers to how to use the framework activities in school as well as using their own knowledge and ideas to implement better FMS. 

The fifth section provides information to help teachers prepare to help children set goals and to subsequently assess their improvements in FMS and PA as a whole group of children and individually. These sections were strongly informed by the voice of the parents (FG1P) and researchers (FG1R), who commented on the need for assessment to be implemented into the existing school reporting framework. Finally, the sixth section lays out ideas for setting homework tasks for children with the consideration of the home environment. This section also details how to engage parents in aiding the success of the programme. Previous qualitative work highlighted the differences in home environments, especially those of deprivation and the need for practice activities to be on a small scale with none or limited equipment use [13]. This section focusses on these elements so they can be achieved by as many children as possible. These two sections support the delivery of the third training session, which focusses on the overall delivery of the programme, the measures for success, and how to create ‘homework’ for children and parents. The remaining materials from Table 8 would also be produced around the central theme, ensuring continuity across materials and ‘branding’.

#### 2.5.4. Task 4.4: Produce, Pre-Test, and Refine Materials

The materials drafted in Task 4.3 were produced as prototypes (rapid experiments to test ideas quickly, simply, and at low cost) for this programme of study. This aided gathering data to validate ideas proposed before from researchers, the literature, and the planning group. Future research should engage in the development of all programme materials with the required experts to produce them, including media promotions and parent packs. These were not developed in this study due to the constraints of this being a PhD project. 

Within the process of IM, once the materials have been produced, they should be pre-tested with the target population of the interventions, and the results of these pre-tests should be evaluated. Pre-testing should include the involvement of the planning group, particularly the intervention end users/participants (children), and those delivering it (teachers). Testing the structure and organisation of the programme and individual intervention applications and components reveals any issues or ineffective elements but also strengths, which can help establish the most effective protocol and materials when identified. Using planning group feedback and questions such as ‘does this method interest you?’ and ‘would this protocol/delivery suit your setting?’ embraces stakeholder involvement and ownership. Within these testing periods, it is key to observe the thoughts around sustainability of the interventions, such as the regularity of delivery and the potential for other factors/determinants within the environment to disrupt this. The evaluation methods should be well-designed to fit within the pre-test delivery rather than as a standalone element. 

In this study, there was scope to pre-test one element of the intervention materials with the planning group. By using Section 3 from the framework booklet, a prospective session was planned using several of the activities presented in the booklet (Appendix A); this was pre-tested with two classes of EYFS children from the planning group. Class teachers were also invited to be part of the session. As part of intervention refinement, children were asked to provide formative feedback; the session leader asked the children if the session was ‘really good fun’, ‘okay fun’, or ‘not so fun’. Across the two classes, 67% of the children found the trial session ‘really good fun’, whereas 10% said it was ‘okay fun’, and the remaining 23% said it was ‘not so fun’. The children were then asked what would have made the session better or what they really enjoyed. Many children commented on the use of equipment as fun, whereas other children wanted to play more sports-based games to make the session more exciting and structured. Teachers provided comments throughout the session to the session leader, with a clear theme that the session would work for some children, but other children require more rules and structure to their activity. Using this formative feedback from the children and teachers, further refinement of programmes materials was performed. The remaining materials drafted and produced in Task 4.3 were not pre-tested in this study due to time and resource constraints, but this opens the opportunity for future research and intervention development. Table 8 shows the pre-test plan for all of the materials from Table 7.

### 2.6. Step 5: Programme Implementation Plan

Step five comprised four tasks. First, we identified the potential adopters of implementation for the programme, which was followed by stating the outcomes and performance objectives for the programme’s use by the adoptees. The creation of matrices of change objectives for use with the programme and designing the implementation interventions to be used were the final two tasks. 

#### 2.6.1. Task 5.1: Identify Potential Adopters of Implementation for the Programme

Step 5 requires the key stakeholders for intervention success (effectiveness) to be identified. Using the planning group’s previous inputs within the initial focus groups (FG1P, FG1R, FG1C) and previous interviews [13], the implementers, adopters, and maintainers of the programme were identified. 

Implementers are considered to be those who will put the interventions within the programme into practice; in this case, they are those who will implement teacher training and framework intervention delivery. For interventions to be successfully adopted, they should be specific to the setting of adopters of the programme, such as schools, clubs, and local authorities. Finally, the implementers and adopters will work together to become maintainers of the programme of change, especially where positive changes are observed, leading to the specified programme objectives and outcomes. 

Within this programme, four key stakeholders were identified:**Implementers:** Local authorities, delivery partners**Adopters:** Teachers, School senior leadership teams (SLT)**Maintainers**: Local authorities, delivery partners, teachers, School SLT

Stakeholders may not exclusively have a singular role in intervention delivery, as mentioned in the list. Local authorities must engage with schools and leadership teams to get them on board with the intervention and to train educators to use the intervention/programme, making them implementers. Consequently, the local authorities must have a method of disseminating the training to educators, and this is via delivery partners. Delivery partners may have different roles in different local authorities but are likely to be public health staff, health visitors, or PA and sports-based professionals. Importantly, it is not specified who this would be, giving authorities the freedom to choose, which could ultimately result in better sustainability. These delivery partners must ensure the maintenance of a relationship with a school and of the training intervention. The adopters of the programme are those based within the school setting, hosting and delivering the programme. Teachers are the ‘frontline’ staff to this intervention by delivering it to children, aided with the support of their school SLT. These teachers must be trained by the implementers. Finally, all roles come together to be the maintainers of the programme. A local authority must commit to keeping delivery partners available for the training of teachers. School SLT and teachers must commit to continuing to deliver the programme to future classes to help change social norms and beliefs while increasing FMS mastery in EYFS children through the programme. 

Considerations to the maintenance of the programme, such as funding and higher-level policy, to ensure time and sufficient structure is in place to allow for the interventions are pivotal. Therefore, members of the school SLT, local authority, and national government should be considered to be key maintainers of the programme and influence later iterations of the IM process. 

#### 2.6.2. Task 5.2: State Outcomes and Performance Objectives for Programme Use

For the implementers, adopters, and maintainers, the main outcomes were stated for the dissemination, implementation, adoption, and maintenance of the interventions. These outcomes were then split down into performance objectives in a similar way to the tasks in Step 2. These were informed by questions used within the focus groups, with a focus on the sustainability of the programmes and their implementation. The outcome and performance objectives are stated in Table 9.

#### 2.6.3. Task 5.3: Create Matrices of Change Objectives for Programme Use

Using these outcomes and performance objectives for the programme, further matrices of change (as in Step 2) were produced (Appendix A) to provide the implementers, adopters, and maintainers with change objectives to aim for, making the programme tangible, implementable, meaningful, and measurable. The proposal for the assessment of these objectives was made within the evaluation plan in Step 6.

#### 2.6.4. Task 5.4: Design Implementation Interventions

In Table 10, the change objectives of dissemination, adoption, implementation, and maintenance are supported by theoretical methods and practical applications, as in Step 3. The delivery channels used within these applications were also considered. These include mediated channels, such as local authority communications with schools (monthly newsletters, meetings, conferences, emails), using videos on social media to promote the intervention, and a website explaining the intervention and how to sign up, and interpersonal channels such as local authority communications with schools via health workers/visitors. Some of the maintenance objectives link closely with the main training and framework interventions. For example, a teacher must evaluate the programme at the end of the year to ensure that sustainable maintenance in their school is achieved. Although this action is a key part of maintenance of the programme, the information for teacher evaluation would be provided in the training and framework documentation seen in the two main interventions. Many of the theoretical methods suggested support the method of facilitation by providing local authorities and teachers with materials to help them achieve adoption, implementation, and maintenance. 

### 2.7. Step 6: Programme Evaluation Plan

Finally, within Step 6 (Figure 1), a plan to evaluate the whole programme and the effectiveness of individual interventions was produced through four tasks. The effect evaluation was planned to establish if the programme of interventions had the desired effect on the target population(s) (teachers, and children). The process evaluation examines how the intervention was implemented and adopted in the desired settings (schools), and it identifies the key implementation characteristics, answering the question of if the desired effect on the population was or was not achieved, and helping to uncover issues within the interventions.

#### 2.7.1. Task 6.1: Process and Effect Evaluation Questions

The first task of the evaluation plan was to write effect and process evaluation questions for the programme. 

##### Effect Questions

Has the programme improved children’s physical self-efficacy and academic performance?How much does the PA level and FMS mastery of the children completing the programme change from pre- to post-intervention?What was the impact of the programme on teachers’ knowledge and self-efficacy to teach and plan for FMS at the EYFS?What was the impact on the children’s knowledge and enjoyment of FMS?What was the structure of FMS delivery like in the participating schools before the intervention?

##### Process Questions

What parts of the intervention worked well, and why? What did not work as well regarding the implementation, and why?What elements of the interventions have been sustained post-intervention?What aided dissemination and adoption of the programme?How often are teachers planning and using the framework in schools?If schools continue to use the programme, why?Have the participants (children) enjoyed the delivery of the intervention in schools?

#### 2.7.2. Task 6.2: Indicators and Measures for Assessment, Task 6.3: Evaluation Methods, and Task 6.4: Evaluation Execution

The final three tasks of Step 6 were collapsed together to achieve a more concise nature of work. With effect and process questions established, the identification of the indicators and measurements for these variables were chosen (Table 11 and Appendix A), informed by the focus group discussions (FG1P and FG1R, [13]). The planning group was asked to suggest important ‘real-world’ outcomes to the programme of interventions that effect the user, implementers, and adopters (Appendix A). Finally, the methods of evaluation were chosen (qualitative and quantitative), and the proposed plan for effect and process evaluations was created (Table 11). The plans were designed to be easy to follow and implement in the school setting, with the activity designed for participants (children), implementers, adopters, and maintainers (local authority and teachers) to complete. Given the importance of the role of theory in helping inform practice, the process evaluation was guided by using the RE-AIM framework [50], which has been used to evaluate the impact of varied public health interventions for over 20 years [51]. The RE-AIM framework aims to support the development of multi-level intervention at the individual, environmental, and policy levels, and it uses the five dimensions of reach, efficacy, adoption, implementation, and maintenance. In this study, the authors proposed to complete evaluation in each of the following areas: reach of interventions, effectiveness of the interventions, adoption of the interventions in the appropriate and targeted settings, implementation of the interventions, and the proposed maintenance of the interventions.

Based on planning group consultation, the literature, and the researcher’s own knowledge, the chosen evaluation methods for this programme of interventions consist of:Card sort activities (Appendix A)SurveysInterviewsObservationsDevice-based measures (accelerometery)

Each of these methods may capture more than one element of evaluation, which can be observed in Table 11. 

## 3. Discussion and Summary

This study was the first to begin to plan an FMS intervention for early childhood populations using IM by iteratively planning and developing interventions in collaboration with key stakeholders. The authors appreciate that not all steps of the IM process were completed, but they were given consideration and recommendation for further work to be completed. This work is important for advancing knowledge of what really matters within FMS and PA interventions for children, and this work goes beyond the traditional approach of many interventions where only Step 4 of the principles of IM (programme production) are adopted for their organisation within reporting. The many tools and tasks involved in the IM process were operationalised within this article to ensure new learning emerges; this knowledge is essential for the future planning of programmes of intervention. The key findings show the thorough and rigorous process of the IM-enabled development of a practical and feasible plan for future intervention for the early years and their specific health behaviours. This section will summarise the emerging outcomes from the current IM process with a socioecological focus, highlighting areas for future development through the following structure:What is the problem?What can be changed?How can it be changed?What is the design of change?Who needs to be involved in change and how?How can change be evaluated?

### 3.1. What Is the Problem?

This study collated evidence from two previous studies by the authors [13,24], unpublished empirical evidence, and existing literature to identify that children are underachieving in their FMS proficiency at the age of 4–5 years old, and by using previous knowledge from IM [12] and a planning group, the behaviours and determinants leading to this issue were identified. Importantly, as evidenced in previous qualitative work [13], children have a lack of opportunity to develop FMS, both at home and also at school, where educators’ practices are not well-structured, guided, or informed by their own continued professional development or training. This is underpinned by a lack of knowledge, lack of self-efficacy for delivering FMS, and the expected social norms of FMS and physical development teaching for the EYFS [13,32,48,53]. These interpersonal and community-level influences shown in the SEM (Figure 2) play a crucial role in enhancing a child’s opportunity and environment to progress their FMS proficiency. On the other hand, children have low FMS competency, which is likely to be personally determined by their enjoyment of PA [54], self-efficacy in their movement ability [45], and knowledge of FMS. Therefore, the current landscape at an educational/policy and personal level leaves children lacking in FMS proficiency, leading to insufficient levels of PA as they age and potentially poor health outcomes in childhood and adulthood. 

### 3.2. What Can Be Changed?

The use of a systems science approach, including creating a causal systems map of how this intervention is to be implemented beyond the school level, should be established [55]. This approach is similar to the 12 local delivery pilots delivered by Sport England [56], promoting a whole systems approach by using local places and people to deliver more PA while understanding the barriers and determinants to people getting or remaining active. The pilots promote the inclusion of local people as key stakeholders in addition to reflecting, testing, and learning from the processes that they put in place, reflecting key IM principles.

Within the current programme of intervention, the individual, their environment, and the environmental agents within it (interpersonal-, community-, and policy-level) need to be changed to provide a multi-level approach to intervention. Addressing this problem by creating supportive environments for children to practice FMS within is essential. The majority of children in England attend school at ages 4–5 years old [57], meaning there is a community-level opportunity to intervene due to their contact with children and ability to train educators within these settings. This identifies that for the current IM process, the targeting of teachers within the EYFS of primary schools in England is crucial. This includes targeting critical determinants, including teachers’ knowledge, self-efficacy, social norms, and their outcome expectations of the intervention, as explored in previous interviews [13]. Individual behaviours were considered important within this intervention, despite the children’s young age and lack of autonomy over the choice of activities in their day-to-day life. Like the environmental agents, the determinants of the children’s behaviours, such as children’s enjoyment of FMS, knowledge of what FMS performance and activities are, and their self-efficacy of their FMS and physical performance, are important to think about when aiming for change for this population group. These determinants were understood by engaging with key stakeholders.

Although this IM process focussed on teacher-level intervention, the researchers felt there would also be a need for an intervention at the senior leadership team (SLT; headteachers, governors, deputies) level (organisational level), which reflects the IM principle of multi-level influences and stakeholder involvement. These individuals may not implement the intervention but are influential in decisions that facilitate engagement with the intervention. The intervention at the SLT level would be similar in terms of information to influence teacher training programmes; however, this would be based around the key benefits of improving children’s FMS and how to promote this at the early years and as a whole school. The primary PE and sport premium funding [31] provides funds for schools to focus on implementing this kind of provision and intervention for staff and children. Therefore, developments around the specific use of funding [32] and how best to support the EYFS staff in the delivery of the intervention proposed in this programme would be key factors. 

Parental attitudes and beliefs around school provision should also be considered an important determinant of the environment to target. Parental attitudes are particularly influential on their child’s behaviours, including their motivation to perform PA [22,58], and thus, they may be influential on their attitudes to PA and PE in the school setting. It is important to consider socialising agents such as the home environment and family-level intervention in future work [21,59]. The focus group discussions conducted with the planning group identified ‘bringing the intervention into the home’ during future implementation or holding interventions exclusively outside of school settings. A child’s parents’/carers’ actions are important influencers in their choices and behaviours, and they want ‘to get support to bring it into the home’. Literature has explored the reduction of childhood obesity in the home setting by using IM [38,39], indicating the use of FMS interventions outside educational settings to be a possibility, although harder to use to intervene due to the broad variation of domains within individual homes. Socioecological approaches use a holistic lens that identifies the influential determinants across the layers of the model, recognising the levels of influence on an individual [37]. Therefore, future IM processes should strongly consider how to positively influence FMS environments outside of school settings, such as the home. Consideration of the variation in SES would be pivotal within future programmes, as children from lower SES backgrounds reportedly spend less time doing PA [30]; therefore, it is likely that these children spend less time practicing FMS. These children are also more likely to be overweight or obese [60], meaning that the need to improve their physical competency through such programmes will be pivotal to their future health and heath behaviours.

### 3.3. How Can It Be Changed?

When identifying the determinants of behaviours and the environment, there is a clear vision to what needs to be changed for the individual and their environment. By establishing broad programme goals, behavioural and environmental outcomes, and performance objectives and change objectives by giving a broad-to-narrow and specific approach, intervention methods and applications can be planned (Table 4 and Table 5).

It is easy to say ‘the self-efficacy of teachers to provide FMS specific teaching needs to increase’, but how can this process be achieved? This is where small and manageable change objectives should be established. As seen in the Matrices of Change (Appendix A), it can be ensured that teachers improve their own self-efficacy by identifying ways for them to understand how to identify improvements needed in the FMS of children, giving them the appropriate means of planning sessions and more effective ways they can engage their children in FMS practice. 

Likewise, for children, it is known that their mastery level of FMS needs to be improved, but what are the potentially effective methods for achieving this within the proposed intervention? By increasing the time children have for PA at school, there may be greater improvements in their FMS [61,62] when partnered with setting goals that are FMS-focussed while providing better structured environments for FMS practice through the target changes for the environmental agent (teacher). 

What has emerged from this IM process is the need to identify the most influential determinants of the behaviours and environments from multiple sources of knowledge, the literature, and the planning group. As the first IM study to do so for this age group, it should be considered that the most influential determinants are actually identified in future IM iterations, and that some determinants may be considered to be less important in future iterations.

### 3.4. What Is the Design of Change?

Intervention design can take many forms, and IM processes target multiple behaviour change theories [49] to address several intervention areas as important. In effect, several small interventions to create behaviour change was proposed as part of a programme (Table 4 and Table 5) and helps to suggest multiple ways to intervene with the end users (children), the implementers (teachers), and their different determinants using practical applications. A key outcome from this study is the preparation of a framework for teachers to use to directly target the change in children’s behaviours. The framework helps to provide guidance, support, and structure, and it simultaneously enhances autonomy and ownership of the intervention [63] for the teacher and children. A framework must be supported by the enhancement of the educator’s knowledge, self-efficacy, and expectations of using a framework delivery. Therefore, the supporting intervention includes the training of teachers for framework delivery in school settings [64]. The FMS School Project Booklet provides a section of activities focussed on physically improving children’s FMS. Using previous literature and intervention techniques to inform this section was important. Two key elements used in the booklet include que words, which were successfully used by Foweather et al. [65] to improve children’s FMS performance. Additionally, the STEP model has been widely used in coaching and educational practices to allow for inclusive teaching [66]. This section also approaches the use of different pedagogies in the FMS sessions by giving linear and non-linear examples for teachers to choose and use [67].

Intervention design should be pragmatic and realistic to the users and adopters, ensuring the intervention can be effectively implemented in the desired setting with the target population [12]. The intervention is designed to ensure teacher burdens and workloads will be minimally increased when implemented, which was identified as important in the previous literature [13] and the planning group. The materials that support and make up fundamental elements of the interventions proposed in this programme have been designed to (a) support the training delivery within this intervention (recapping knowledge taught and provided in these sessions) and (b) be quick and easy to use in educational planning, delivery, and evaluation of the programme by providing pre-made materials. 

### 3.5. Who Needs to Be Involved in Change and How?

Intervention requires the collaborative efforts of many stakeholders to improve the possibility of success, as highlighted by the systems science approach [55]. This programme focusses on improving the provisions in schools by using teachers as adopters and maintainers of the intervention, with children as the end users with the desire to change their PA and FMS behaviours. However, other key stakeholders include but are not limited to (from intrapersonal-level to organisation- and policy-level stakeholders):Parents—although parents have no direct role to play within the current intervention and its delivery, they can enhance the success and long-term outcomes achieved during the intervention period by engaging with the resources provided to them during the intervention. Engaging directly with parents is at the discretion of each individual school, and as with all interventions, it will have varying degrees of success. Despite this, communication with parents and the role they can play should be considered to be critical, especially in future work.School SLT—these key stakeholders must agree that using the intervention, training their staff, and use of the framework intervention is sustainable and worthwhile within their school.Local authorities—they are key partners in the implementing the programme in schools. Without local authorities, there is no initial platform to deliver and communicate the intervention from. Their influence on schools within the local authority should be key in ensuring sufficient and successful intervention uptake. In the long-term, which is dependent on intervention success and stakeholder opinion, local authorities could stipulate a mandatory need for the intervention in schools at the EYFS level.Delivery partners—they are key players within the local authority set-up, as they provide the training sessions for the intervention to teachers. They should support schools and teachers beyond the training, ensuring successful implementation within school environments.Public—policy makers at the government level represent the highest and possibly most influential level this intervention could reach: affecting public policy. The requirement for a statutory FMS intervention or improved framework at the EYFS could be pivotal to ensuring healthier and more active lifestyles for children from an early age. This programme could be delivered locally but be evaluated at a national level, much in the way children are assessed in literacy and maths skills.

### 3.6. How Can Change Be Evaluated?

When evaluating the changes achieved by the intervention, the process and its effects should be considered. As shown in Step 6, curating questions, followed by planning the use of practical techniques, is important for evaluation implementation. Using the RE-AIM framework [50] and evaluating the reach, efficacy, adoption, implementation, and maintenance of the intervention at the individual and community levels helps to focus the evaluation. Future work should also use this framework at the policy level. Working with government policymakers to implement policy changes for the early years and EYFS curricula in relation to more specific FMS tuition, guidance, and practice for this age group would be a critical policy change. The reach of the policy change and effects at the delivery level (EYFS settings); the efficacy of the policy change for the teachers and children involved (evaluating whether it achieved what it set out to: increase teacher knowledge, delivery, and confidence; increase children’s FMS competency and likelihood to continue with good PA habits); the adoption of the policy by local authorities, schools, and educational settings; the implementation of what the policy prescribes or recommends; the maintenance of policy use at the government level, local authority level, and school level will be essential to constructive evaluation processes. 

This IM plan developed the effect evaluation around the most important variables to be changed during the intervention, according to the environment, individual behaviour, and their determinants:Environmental outcome: teachers planning to teach FMSBehavioural outcome: children practicing FMSDeterminants: improving teachers, parents, and children’s knowledge of FMS, increasing the self-efficacy of the teachers to deliver FMS content/activity in school settings, providing children with an enjoyable intervention/FMS practice

Future intervention should consider if the intervention approach being used targets the elements listed previously while considering how these may be evaluated. 

During the process evaluation, attention should be paid to how the intervention is delivered during implementation and in practice. This evaluation process should provide important information about:The completeness of delivery: was the programme delivered as intended with all its elements, and if not, why?Continuation of the intervention: once implemented in the school setting, was it continued successfully and appreciated by the adopters, maintainers, and users of the programme?Participant exposure: did the participants of the intervention receive the appropriate dose of the intervention?

These elements of process evaluation may help provide important answers as to why the intervention did or did not work [68]. The methods for evaluation at the process and effect levels should be pragmatic and realistic; this intervention avoided using overly scientific measures of progress, as these are unrealistic in school settings. Observations, questionnaires, and conversations with key stakeholders of the intervention, including the use of card sorting activities, were deemed to be important in these processes. 

### 3.7. Strengths and Limitations

When examining the strengths of the current study, to the authors’ knowledge it is the first to use IM to plan an FMS intervention for EYFS children within school settings, thereby providing a high level of novelty. This means that the processes throughout were strongly rooted in the use of the socioecological model and engagement with a planning group throughout that included stakeholders from varying levels of the model. The study brought together researcher knowledge, existing literature, and stakeholder engagement to address their wants and needs.

Reflecting on the limitations of the current study, many of the materials proposed in this study have not been drafted, produced, or pre-tested with the planning group or stakeholders. This also represents a need to engage with the appropriate producers to establish and create these materials. Despite the current planning group in this study, there is certainly still scope to engage with other key members, including local authorities, third sector organisations and charities, and the national government. Due to local authorities being identified as implementers in the current IM process, their involvement in future development and research should be considered pivotal. A further limitation of this study is the small sample size used with the current planning group and the use of convenience sampling. This limits the cultural, socioeconomic, and geographical transferability of the results within this study. However, the authors recommend the drafting, production, and pre-testing of materials with larger planning groups across a varied socioeconomical, cultural, and geographical sample to improve these outcomes. Engaging with planning groups representative of communities across England would be important in future work.

## 4. Conclusions

This study aimed to provide an important and well-informed basis for the development of future interventions. By considering the socioecological model when designing the current programme of intervention, the researcher’s attention was focussed on the individual and the influence of their interpersonal environment, organisational environment, community environment, and policy environment on their behaviours and health outcomes. The identification of these influences and knowledge built within this study as well as in previous research has played a key role in the aims of the intervention and the outcomes to be achieved. This IM study shows that initial intervention may aim to aid children and their interpersonal and organisational environment at the school and teacher levels to provide better structure and opportunity to develop FMS, which may lead to improved levels of PA and positive changes in health outcomes.

## Figures and Tables

**Figure 1 children-10-01004-f001:**
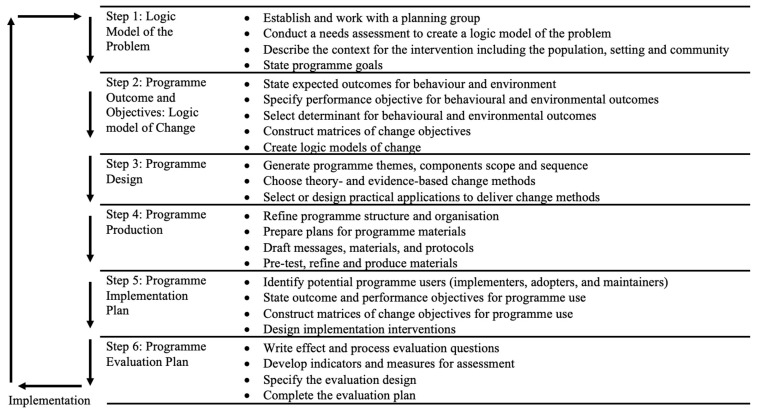
Six Steps of Intervention Mapping.

**Figure 2 children-10-01004-f002:**
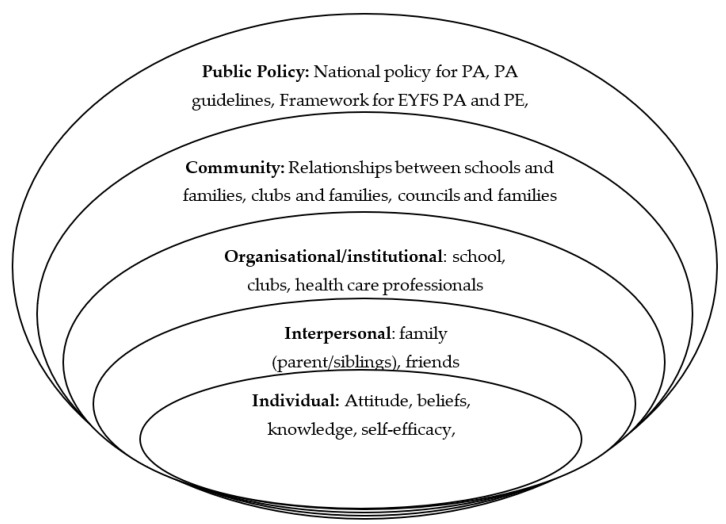
Socioecological Model according to FMS competency and PA in early childhood.

**Figure 3 children-10-01004-f003:**
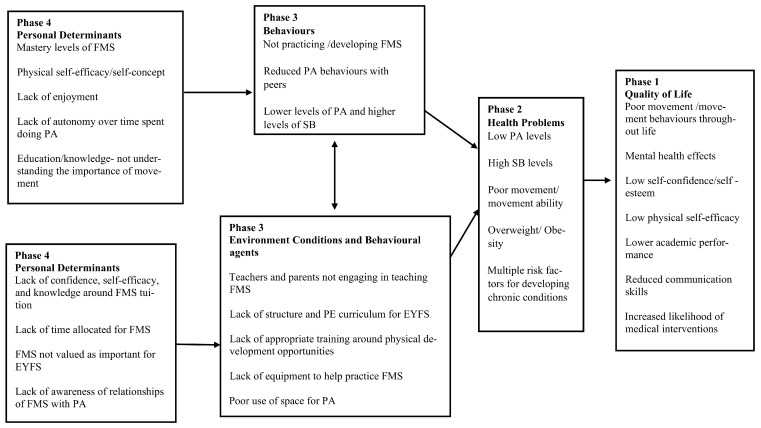
Task 1.2: Logic Model of the Problem. FMS = fundamental movement skills, PA = physical activity, SB = sedentary behaviour, PE = physical education, EYFS = early years foundation stage.

**Figure 4 children-10-01004-f004:**
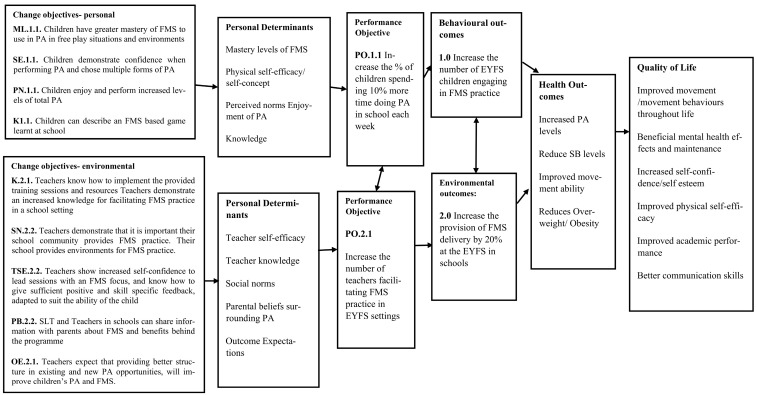
Logic model of change 1. FMS = fundamental movement skills, PA = physical activity, SB = sedentary behaviour, PE = physical education, EYFS = early years foundation stage, SLT = senior leadership team.

**Figure 5 children-10-01004-f005:**
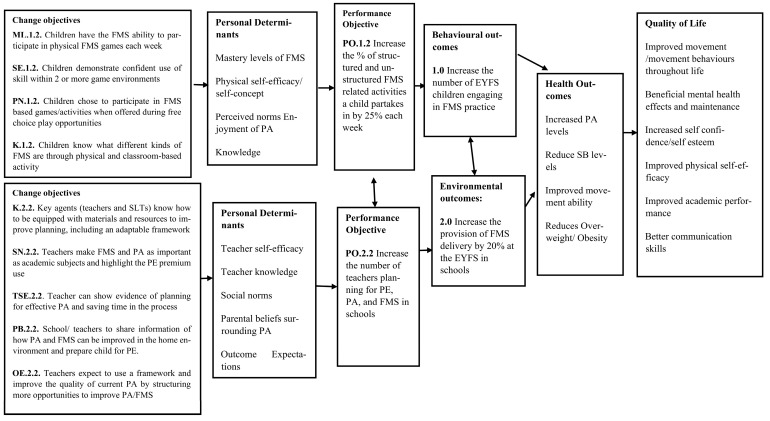
Logic model of change 2.

**Figure 6 children-10-01004-f006:**
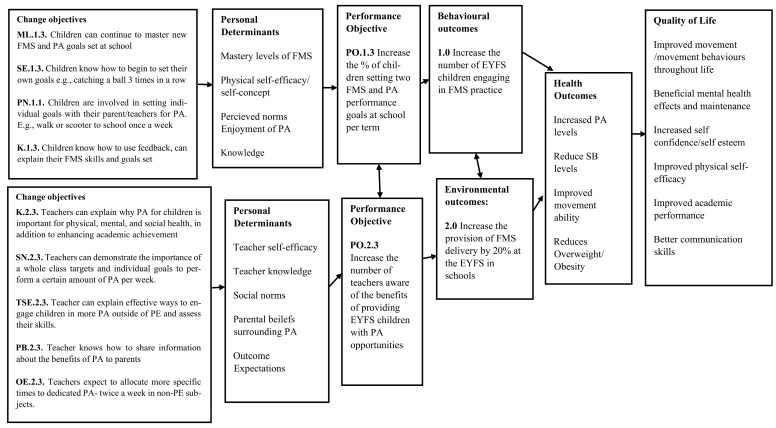
Logic model of change 3.

**Table 1 children-10-01004-t001:** Task 1.2: Key Questions.

Problem	Low FMS Competency in the Early Years
**Whose problem is it?**	Children, parents, teachers, school senior leadership, community provision, government
**Who is the problem affecting?**	Young children
**What behaviours are causing or are related to the problem?**	Sedentary lifestyles, poor-quality PA, lack of planning and training, lack of knowledge and time to enhance this
**The environmental conditions causing or related to the problem**	Reduced outdoor space, poor PA facilities, incorrect use of resources, lack of time for PA, lack of training for teachers
**Determinants of these behaviours and environment**	Underfunding in communities and schools, poverty, lack of practitioner knowledge of FMS, cultural norms associated with PA

FMS = fundamental movement skills, PA = physical activity.

**Table 2 children-10-01004-t002:** Task 1.3 Asset assessment.

Social Environment Asset Assessment	Information Environment Asset Assessment	Policy/Practice Environment Asset Assessment	Physical Environment Asset Assessment
a. Primary schools	a. School newsletters/bulletins	a. EYFS framework	a. School spaces
b. Community sports groups	b. Word of mouth	b. PE curriculum	b. Local community spaces/parks
c. Parent groups	c. Local authority communications	c. CMO activity guidelines	c. Local sports teams/leisure facilities
d. Youth groups/after-school clubs	d. Social media channels	d. Teacher training for early years	d. Home environments

EYFS = early years foundation stage, CMO = chief medical officer, PE = physical education.

**Table 3 children-10-01004-t003:** Intervention components, scope, and sequence.

Intervention Components:	Intervention Scope:
**Intervention one:**	**Programme delivery by delivery partners to teachers:**
Teacher training Three sessions of training delivered by delivery partners Theory and practical elements of learning Evaluation of training (see Step 6)	Delivered across three 2 h sessions Made up of theory elements and active practical elements Supports the framework documentation Evaluation of training delivery (see Step 6)
**Intervention two:**	**Programme delivery by teachers in the school setting:**
Framework documentation Information on FMS, PA, and health Physically active sessions split by domain Classroom sessions Planning templates Homework ideas Promotion techniques in schools How to report on a child’s FMS How to evaluate the intervention Parental packs and education	Delivered across a whole school year of EYFS aged 4–5 years old Daily activities incorporated Reports of child achievements within each school report/end of term Teacher evaluation of programme at end of each school year Delivery of parent materials across three school terms

FMS = fundamental movement skills, PA = physical activity, EYFS = early years foundation stage.

**Table 4 children-10-01004-t004:** Behaviour change and practical applications from Step 3: Task 3.3 (basic methods).

Determinants of the Problem	Behaviour Change Methods	Parameters	Practical Applications and How Delivered
**Individual**			
The social norm is for children to be sedentary	Change the perceived norms: Belief selection Role modelling	Knowledge of children’s existing beliefs and simple messaging for children are required Attention shown to role modelling, self-efficacy to be active/use skills	Promotion in school settings (posters, children’s work on display), have activity-based lessons Role modelling by teachers to be active with the children and provide them with increased active opportunities
Children lack knowledge about their fundamental movement skills	Changes in knowledge: Active learning Individualisation	Time bounds of school and lesson time, information available to teachers Responds to each child’s needs	Framework activities: FMS-based and themed activities within school learning opportunities (practical and classroom-based) STEP model
Children lack mastery to perform good FMS and increase their PA	Increase children’s skills: Active learning Tailoring	Skills of teachers to improve child mastery, time in school Matching children’s existing skill levels	Framework activities: practical activities and planned sessions (organisation) Teacher-planned sessions using STEP
Children lack self-efficacy in their movement	Increase support and goal setting: Feedback	Needs to be individual to the child and specific to their skills	Teacher to work one-on-one with the children in class Set goals with children each term
**Environmental**			
Teachers lack self-efficacy and knowledge to teach and provide FMS in education setting	Increase teachers’ skills: Active learning Changes in knowledge: Individualisation Feedback	Existing skills and knowledge, information available in training sessions Responding to each teacher’s needs Specific to the teacher and their school at a given time	Provide interactive training content for teachers to engage with (multiple sessions) Framework provides opportunity to personalise session delivery to each teacher’s skills Delivery partner to review progression with framework each term
Parents’ lack of awareness to help children practice skills taught at school at home	Increase awareness: Persuasive communication Facilitation	Relevant messages to the parents and their children Surprise and repetition Identifying barriers to participation	Advertising the intervention happening in school, send home progress in monthly newsletter Use as homework tasks for children to do at home with their parents
School settings and environmental agents lack knowledge to provide FMS education at the EYFS and do not know how it links to PA	Changes in knowledge: Active learning Tailoring	Time available to provide new knowledge Matching to the culture of school and/or SES	Videos within training for teachers to use widely in school Train teachers to match the needs of children and school (number of children, skills of children, etc.)
Teachers and schools feel it is the social norm not to be aware of FMS education and practice for EYFS children	Change the perceived norms: Belief selection	Knowledge of teacher’s/school’s existing beliefs of PA and FMS	Role modelling (delivery partners) Testimonials of other schools using the intervention successfully

FMS = fundamental movement skills, PA = physical activity, EYFS = early years foundation stage, STEP = space, task, equipment, people, SES = socioeconomic status.

**Table 5 children-10-01004-t005:** Detailed change methods and applications for programmes of intervention, Step 3: Task 3.3.

Broad Practical Application and Basic Change Method	Detailed Change Method	Detailed Practical Application	Population, Context, Parameters
Framework and increasing skills (active learning)—children	Guided practice—repeating behaviours several times with feedback Setting graded tasks—increasing the difficulty of task as behaviour improves Stimulus control—adding more cues for healthier behaviours	Practical session delivery from the framework Using the STEP model to grade the tasks in PA sessions Adding more cues to be active within school day	*Population:* Children *Context:* In-school PE lessons/active times *Parameters*: Different sessions can be planned from the framework, time available to deliver the sessions
Framework and increasing knowledge (active learning)—children	Using imagery: using artefacts with a similar appearance to the subject Chunking—stimulus patterns to make parts of a movement a whole	Using images of skills within classroom-based activities or images so that children begin to associate with the skills (e.g., animals) Splitting skills down by component parts through the activities performed	*Population:* Children *Context*: In-school classroom lessons *Parameters:* Familiarity of the images/activity to the children
One-to-one with children and increasing support for self-efficacy	Goal setting—prompting planning to reach goal-directed behaviours Providing cues—consistent cues throughout sessions	Teachers set goals with children—example goals within the framework Children are given opportunities to develop their own cues to use when performing skills	*Population:* Children *Context:* In-school PE lessons/active times *Parameters:* Children best to develop their own cues for performance
Training and increasing skills (active learning)/increasing self-efficacy—teachers	Guided practice—repeating behaviours several times with feedback Mobilising social support—instrumental and emotional social support for teachers Planning coping responses—identifying barriers and ways to overcome them	Teachers rehearse using framework and gain feedback from delivery partner during and after delivery using own self-evaluation Teachers identify barriers within their classes within training and formulate plans to overcome these	*Population:* Teachers *Context:* Training session and regular contact with delivery partner *Parameters:* How much support a delivery partner can give
Training and changes in knowledge—teachers	Advance organisers—presenting an overview of the material Discussion—encourage debate over a topic Using imagery—using artefacts with a similar appearance to the subject	Present an overview of the framework within the training for the teachers (over three sessions) Hold discussion within training sessions to discuss ideas proposed Videos demonstrating how skills can be performed, videos of activities with children	*Population:* Teachers (and children—imagery) *Context:* Training session *Parameters:* Length of training sessions, available resources, prior knowledge for discussions
Changing social norms of schools	Public commitment—pledging to engage in healthier behaviours Cultural similarity Provide contingent rewards—praising and encouraging behaviours	School publicly says they will be running the programme through newsletters, school displays, within reports Using messages from other schools (preferably local) to show the success of a programme School openly praises healthier behaviours with rewards (more time to be active)	*Population:* Whole school community *Context:* Delivery in school *Parameters:* sociocultural characteristics of specific schools, praise must only follow the specific behaviour
Parent packs and increasing awareness: persuasive communication	Consciousness raising—providing information and feedback Framing—gain-framing messages and the advantages of healthy changes	Homework tasks—provide information about school tasks and reports- feedback for parents to act on Use reports to demonstrate the benefits; parent packs of information and benefits to child and family	*Population:* Parents *Context:* Actions performed at home *Parameters:* Gain frames to be used rather than loss frames to use positive messages; self-efficacy of parent and child to be considered

PA = physical activity, PE = physical education, EYFS = early years foundation stage, STEP = space, task, equipment, people.

**Table 6 children-10-01004-t006:** Programme organisation and delivery channels.

Environmental Agent Intervention	Individual Intervention
**What (intervention):** Teacher training sessions on FMS, PA, health, and framework delivery	**What (intervention):** Adaptable framework delivery in schools
**Who:** Delivery partners (facilitators) and teachers (target group/participants)	**Who:** Teachers (facilitators) and children (target group)
**Where (setting):** In-school settings or local authority settings	**Where (setting):** In schools, in the EYFS
**When (sequence):** Before autumn/winter, spring, and summer term at school (in England) = three times per year	**When (sequence):** Delivery of the framework across a whole school term
**How much (scope):** three training sessions per training block	**How much (scope):** Delivery of an intervention application at least three times per week
**How often (scope):** three times per year	**How often (scope):** Designed to be used for each term of school
**Interpersonal delivery channels:** *Dissemination:* Delivery partner leader to lead the training (with peer leaders)	**Interpersonal delivery channels:** *Awareness:* Teachers discussing with parents, health visitors promoting the intervention to schools, volunteer parents who have previously observed success in the programme *Dissemination:* Teachers working with children, children working with peers, parents aiding children, volunteer parents
**Mediated delivery channels:** *Dissemination:* Written print (framework document), videos of training, social media groups (network of teachers to discuss ideas), flip charts, media presentations, recorded training session archives	**Mediated delivery channels:** *Awareness:* Using videos on social media (school) to promote the intervention, a website explaining the intervention to children and parents, school newsletters and displays within the school, written information sent to parents, texts/app messages sent from school to parents *Dissemination:* Printed materials with information, videos for children to watch, a website showing activities done at school and ones for at home, social media groups for parents to discuss their home activities

**Table 7 children-10-01004-t007:** Task 4.2: Programme material plans.

Programme Component	Description	Producers	Drafted in This Study
Video of the intervention for schools	Video showing how the framework works in action. Images of the training and then of teachers delivering in school settings. Feedback/testimony of previous training experience by SLT and teachers.	Media experts (filming, production, etc.)	No
Newsletter promotions	Small snippets of information about the programme/intervention within school newsletters between staff to catch the attention of other teachers/SLT Small snippets of information about the programme/intervention to demonstrate to parents what schools are doing	Researchers to provide templates for schools to use in their own dissemination	No
Social media promotions	Posts about the intervention programme posted regularly to the intervention’s social media account to inspire teachers with weekly FMS planning with links to the website to sign up for training Posts schools can use to promote the use of the programme in school	Social media experts/graphic designers	No
Websites about intervention	Website with pages with the intervention resources for teachers to access Website with pages of engaging content for parents to use with children and to help with home activities	Website producers	No
Training sessions	Slides accompanied by verbal delivery from a delivery partner, practical practice environments, worksheets to complete during the sessions	Researchers design these materials/plans for delivery in partnership with a group of delivery partners	Yes (one session)
Framework booklet: Information about FMS	Section of booklet reiterating information delivered in the training for teachers to refer to and strengthen their knowledge while delivering the programme/intervention	Printing services, visual/graphic designers	Yes
Framework booklet: Planning practical activities	A section split by FMS domain that provides activities lasting 5–15 min. All activities must have a STEP adaptation example and list equipment needed, time taken, and outcomes from each activity; a planning section for teachers to plan the use of activities in extended PA/PE sessions; homework activities for children to do *STEP model, guided practice*	Printing services, visual/graphic designers, researchers designing activities	Yes
Framework booklet: Planning classroom activities	A section of examples of classroom-based activities that promote FMS or use stimulus to enhance children’s learning; suggested academic areas for activities, including homework activities. *Imagery and stimulus cues*	Printing services, visual/graphic designers, researchers designing activities	Yes
Framework booklet: Activity sheets	Training: sheets for teachers to complete related to the training content to help solidify learning and to make their own notes Framework: goal setting activity sheets for children to complete with their teacher each term to set goals about their FMS *goals setting* Other activity sheets related to other curricular areas that will combine the knowledge of FMS for children (maths, English, art, etc.)	Printing services, visual/graphic design, researchers designing activities	Yes (in framework)

FMS = fundamental movement skills, PA = physical activity, PE = physical education, SLT = senior leadership team, STEP = space, task, equipment, people.

**Table 8 children-10-01004-t008:** Task 4.4: Pre-test plan.

Programme Application/Component	Pre-Test Objectives	Pre-Test Population	Pre-Test Procedure
Video advertising the intervention for schools	Improve awareness of FMS for teachers within training and to refer to within the framework	SLT and teachers	Send link of video to watch and provide feedback on (used in training)
Newsletter promotions	Teachers know how to share information about the FMS programme with parents Teacher can share class targets with whole school and demonstrate a community for FMS practice at school	Teachers and parents	Provide teachers with template newsletter excerpts and ask them to report the ease of editing and the interest reported by parents and staff when shared
Social media promotions	SLT and teachers can easily share information with parents about PA and FMS at home and the benefits of the programme	SLT, teachers, and parents	Schools share the social media posts with parents (on school Facebook/apps). Parents are then asked if they have heard about the intervention (to measure effectiveness)
Websites about intervention	Key agents (teachers and SLT) are equipped with materials and resources to deliver the framework	SLT, teachers	Teachers and SLT are asked to navigate the websites and report the ease of finding materials and information they wanted/needed
Training sessions	Teachers know how to implement the framework within their school setting Teachers have increased knowledge of FMS	Teachers	Conduct pilot sessions of the training with a group of EYFS teachers, with feedback opportunities for the training content
Framework booklet: Information about FMS	Teachers can explain why PA is important for child health and academic achievement	Teachers	Provide teachers who were trained in the pilot sessions with the booklet
Framework booklet: Planning practical activities	Teachers have increased self-confidence to lead FMS sessions with skill-specific adaptations Teacher can plan effectively for FMS Children engage in the FMS activities to improve their mastery and demonstrate higher confidence in their FMS	Teachers and children	Provide teachers who are trained in the pilot sessions with the booklet. Ask them to plan two practical sessions from the booklet to do with their classes and provide feedback on its ease of use to plan teacher sessions Children can give formative feedback at the end of the sessions (if they enjoyed them, if they were fun, etc.)
Framework booklet: Planning classroom activities	Teachers can use framework to allocate more FMS-based activities to improve PA/FMS Teachers make FMS and PA as important as other academic subjects Children know about different kinds of FMS and can describe an FMS game from school	Teachers and children	Provide teachers who are trained in the pilot sessions with the booklet. Teachers can plan to use classroom activities twice a week with their classes and provide feedback on the success in other curriculum areas Children are tasked with homework to do a game/activity at home with their parents to show what they have learnt/understood from the programme
Parent packs/information	SLT and teachers can easily share information with parents about PA and FMS at home	SLT, teachers, and parents	Teachers are asked to share the information packs with parents when homework is set for the children. Parents are asked if the information helped them to complete the homework with their children (Likert scale, e.g., helped a lot, somewhat helped, did not help nor hinder, did not help, very unhelpful)
Framework booklet: Activity sheets	Children are involved in setting PA and FMS goals and can begin to set their own goals Teacher can demonstrate the importance of FMS targets for children	Teachers and children	Provide teachers who are trained in the pilot sessions with the booklet. Teachers can plan to use the goal-setting sheets with the children once a term and provide feedback on its ease of use and suitability.

EYFS = early years foundation stage, PA = physical activity, SLT = senior leadership team, FMS = fundamental movement skills.

**Table 9 children-10-01004-t009:** Outcomes and performance objectives for programme use.

Outcome for programme dissemination/implementer (local authority)
**3.0** Increase the number of delivery partners delivering The FMS School Project training
**PO.3.1** Increase the number of schools receiving The FMS School Project information
**PO.3.2** Increase the number of local authorities using delivery partners for The FMS School Project training
**PO.3.3** Increase the number of local authorities planning to use delivery partners for The FMS School Project training
**Outcome for programme implementation and adoption (delivery partner)**
**4.0** Increase the number of schools delivering The FMS School Project framework
**PO.4.1** Increase the number of schools receiving The FMS School Project training
**PO.4.2** Increase the number of schools planning to receive The FMS School Project training
**PO.4.3** Increase the number of schools interested in using The FMS School Project training
**Outcome for programme maintenance (teachers and school SLT)**
**5.0** Increase the number of schools using The FMS School Project framework for more than one school year
**PO.5.1** Increase the number of teachers evaluating The FMS School Project framework at the EYFS
**PO.5.2** Increase the number of SLTs granting the appropriate funds for The FMS School Project
**PO.5.3** Increase the number of schools providing the appropriate time for The FMS School Project framework
**Outcome for programme maintenance (local authority and delivery partners)**
**6.0** Increase the number of delivery partners delivering The FMS School Project training for more than one school year
**PO.6.1** Increase the number of delivery partners/local authorities evaluating The FMS School Project training
**PO.6.2** Increase the number of local authorities granting the appropriate funds for The FMS School Project training delivery
**PO.6.3** Increase the number of local authorities providing the appropriate time for The FMS School Project training

FMS = fundamental movement skills, SLT = senior leadership team, EYFS = early years foundation stage.

**Table 10 children-10-01004-t010:** Task 5.4: Implementation interventions.

Change Objectives	Theoretical Methods	Intervention Applications
**Dissemination (of training)** Disseminate information and benefits clearly about the programme to schools LAs know the scope, sequence, themes, and components of the programme and which delivery partner in LA will be appropriate LA can employ delivery partners and explain why it is normal to use them for this kind of intervention	Persuasive communication Tailoring Individualisation Persuasive communication Tailoring Persuasive communication Discussion	Local authority communication with schools (visits) Advertisement on social media, videos, newsletters, websites Written materials Website with information about intervention and for LA DPs to sign up LA advertising to existing members of staff; focus groups with staff to identify appropriate staff member roles
**Adoption and implementation (of training)** DPs know how to explain the scope, sequence, themes, and components of the programme to teachers and explain why it is good CPD DPs can explain the training and framework of the programme DPs can communicate with schools and SLT and make a case about programme delivery and training	Persuasive communication Individualisation Advance organisers Active learning Modelling Discussion Persuasive communication Tailoring	Local authority communication with schools (visits); advertisement in social media, videos, newsletters, and emails Written training materials Practical training for DPs Practical training for DPs Scenario practice (Communications as previously mentioned)
**Maintenance (of framework)** Teachers can demonstrate the evaluation methods of the programme to identify success and areas of improvement SLTs can explain how to grant the appropriate funds for the intervention and equipment from the PE premium SLT and teachers can explain and use time and timetabling effectively to allow for a whole school year of The FMS School Project	Participation Active learning Facilitation Facilitation Advance organisers Participation Persuasive communication Belief selection	Teacher uses evaluation materials provided in the framework Draft reports for SLT that teacher can populate with outcomes Information for SLT about the use of funding and PE premium use for programme (on paper and on a website) DP visits to schools to help plan funds for delivery SLT see the benefits of the programme within their schools DP visits schools to help plan time for delivery
**Maintenance (of training)** DP and LA know how to evaluate the delivery of the training programme LA can allocate funding for employing DP and the delivery of training LA can allow DPs appropriate time to plan, prepare, and deliver successful training	Participation Active learning Facilitation Participation Persuasive communication Individualisation Facilitation	DP/LA uses evaluation materials provided in the training Draft reports for LA to populate with outcomes LA see the benefits of the programme for schools Guidance documents for LA on allocations of money for delivery of DP and training; adjustments according to previous results (e.g., more or less funding) Guidance documents for LA on allocations for DP planning and delivery time of training

DP = delivery partner, LA = local authority, SLT = senior leadership team, CPD = continued professional development.

**Table 11 children-10-01004-t011:** Step 6: Programme evaluation plan—effect and process.

Variables	Indicators and Time Frame	Methods and Execution
**Effect**		
**Quality of life ** Physical self-efficacy Academic performance	One school term: Physical self-efficacy rated higher Improved academic performance	Questionnaire [52] Teacher assessment/observation
**Health outcomes** Better PA levels Better movement ability	One school year: Weekly PA increases Can complete more complex movement tasks	Accelerometry measurements Class-based assessment—reported in survey
**Behavioural outcomes ** Practicing FMS Increasing PA levels with peers	One school term: Better FMS mastery Spends more time in moderate–vigorous PA	Observational assessments Accelerometry measurements
**Environmental outcomes** Teachers engaging with FMS teaching Structure for EYFS, FMS, and PE	One school term: Frequency of framework use Planning for sessions completed	Survey/tracking Survey/tracking (submission of evidence)
**Determinants of change ** Knowledge (T) Self-efficacy (T) Enjoyment (C) Knowledge (C)	One school year: Can identify FMS domains and activities related to them Has confidence to use framework and plan sessions from it Can name an activity they enjoy completing related to the framework Shows knowledge of different FMS	Interview Interview Card sort Card sort
**Process**		
**Programme implementation** Dissemination Adoption	Number of schools completing The FMS School Project Training Number of schools using The FMS School Project framework in practice for a term	Local authority survey of schools delivered to Survey of schools that have received training
**Implementation** Completeness Fidelity Continuation Programme users’ evaluation Programme users’ barriers	Three training sessions delivered Practical, classroom, and home activities used from framework in schools All elements in training are covered using designed materials One year of use in schools and LAs Enjoyment to deliver, ease to deliver Issues with delivery, barriers to use	Survey numbers from LA Tracking by teachers in schools Observation of training sessions School/LA survey Teachers/SLT/DP/LA—interviews, teacher card sort Teachers/SLT/DP/LA—interviews
**Intervention exposure** Participant exposure (use of materials) Participant evaluation	Number of times framework was delivered per week Child enjoyment, child-identified benefits/feelings	Survey/tracking Card sort

PA = physical activity, FMS = fundamental movement skills, DP = delivery partner, LA = local authority, SLT = senior leadership team, CPD = continued professional development, FMS = fundamental movement skills.

## Data Availability

The data presented in this study are available on request from the corresponding author. The data are not publicly available due to privacy issues.

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
