# Peer review of "Improving Fundamental Movement Skills during Early Childhood: An Intervention Mapping Approach"

_children, 2023, doi:10.3390/children10061004_

Round 1

Reviewer 1 Report

The major problem of this article is: where are the objective evaluation and discussion of the obtained results? Without that, how can you propose the model as useful and practical?

So, we recommend authors to be more operational with results presentation and discussion; and, with the project limitations, namely, those related with sample dimension and cultural, socioeconomic and geographical specificities.

In Figure A4. Task 6.4: Child Process Evaluation Card Sort, photo for "Learning new skills" is controversial: it may interpretated as sexist. Also, the picture focus on a physical, product approach, and not on a qualitative, movement approach.

Author Response

Thank you for your comments and review of our manuscript.

We do not feel there is a need to change the presentation of the results or discussion within this article. This is because this article is an initial step in this project, as mentioned in section 1.3 (page 5). We have added some clarity within the article itself. Please see pages 5 and 43. Our approach does not make objective evaluation necessarily possible at this stage, however, this would be achieved in future work. We feel it is important to publish work at both formative and summative phases.

We have added more detail surrounding the limitations of the article in section 3.1 (page 48). Figure A4. Task 6.4 has been changed according to your concerns.

The other two reviewers of this article commented and congratulated us on the good standard of the work and did not express any concern or need for the content to be amended. We hope that therefore the passages we have added to this work help to make it clearer about the article’s intention, the intervention mapping principles it has followed, its value to future work and development, and its novel contribution to the literature.

Reviewer 2 Report

Dear authors,

I would like to thank you for the effort they have put into the preparation of the manuscript entitled: Improving Fundamental Movement Skills during Early Childhood: An Intervention Mapping Approach. The study was aimed to use intervention mapping to design a programme of school-based intervention to improve fundamental movement skills for children aged 4-5 years old.

The manuscript is well-structured and well-developed, which allows for a good understanding of the design of the intervention programme. Congratulations!

Here are some comments on the manuscript:

1)    More references to intervention mapping should be included in the paragraph on lines 116-137. The information as drafted is appropriate but some references should be included to support it scientifically.

2)    Check whether figures 1 and 2 have been retrieved from any source of information. If they are from a source, the reference should be included; if they are self-authored, the reference does not need to be included.

3)     Revise the text formatting of some sections of the manuscript: first three lines of page 8, page 9 (performance objectives for behavioural outcomes and performance objectives for environmental outcomes), table 3, table 10 and table 11.

4)   The first-person plural is used to describe information in the paper. For example, line 164 reads: "Combining what we know about FMS and PA, interventions during early childhood...". In scientific articles, the third person plural or indirect sentences are more appropriate. For example: "Combining what the authors of this project know about FMS and PA...".

Similarly, the possessive determiner "our" should also be changed. For example, line 168 mentions: "to our knowledge there are no studies...". Instead, the wording could read: "no previous studies have been found...".

The entire manuscript should be revised in this regard.

5)  In many tables, information is repeated. For example, in table 7 "programme component, description, procedures and drafted in this programme" is repeated. It should be assessed whether it is necessary to duplicate this information. Tables 4, 5, 7 and 8 should be revised.

6)   Check whether the numbering of the table on page 9 of the supplementary material is correct. Perhaps instead of being called Appendix A1, it would be table A8. If it is decided to change the numbering, the wording in lines 634 and 635 of the manuscript should be changed.

7)  Line 671 mentions "four key stakeholders", but subsequently only three are numbered and detailed. Review whether it would be appropriate to change the text to "three key stakeholders".

8)      In figure A5 there are two blank cards. Check if this is correct or if there has been an error.

9)      Table 11 mentions that a questionnaire will be used to assess physical self-efficacy. It would be useful to explain exactly which questionnaire will be used. The reference of the questionnaire can be included.

10)   The list of references should be adapted to the requirements of the journal.

Congratulations on your work!

Best regards.

Author Response

Thank you for your thorough and helpful review, we feel this has improved our work and appreciate the time and effort made to improve our manuscript. Please see list of amendments attached.

Best wishes

Reviewer 3 Report

Dear Authors

This study examined to improving fundamental movement skills during early child-hood: an intervention mapping approach. The manuscript has been surely well designed and written. It is very interesting study and it presents good data on this very important topic. I believe that this is very excellent issue in field of children section. Moreover, this topic was not clearly revealed in the field of physical activity section. So, I would like to thank the authors for their work on this manuscript.

Minor concerns

Whole manuscript.

Please, refer to author’s guideline and keep the journal formatting of manuscript.

The reference format has to change from (number) to [number].

Introduction, Methods, Results, and Discussion section.

In general, the Introduction, Methods, Results, and Discussion section are quite good.

References section - You have to change references’ format in whole manuscript.

I recommend that this manuscript should be edited by an English professional editor for more readable. There are some typo and grammatical errors.

Author Response

Thank you for your review and comments. We have made the required edits to the formatting and had the grammar checked to ensure better readability.

Round 2

Reviewer 1 Report

no further comments